# Symptomatology of Fibromyalgia Syndrome in Men: A Mixed-Method Pilot Study

**DOI:** 10.3390/ijerph19031724

**Published:** 2022-02-02

**Authors:** Ilga Ruschak, Loren Toussaint, Lluís Rosselló, Carina Aguilar Martín, José Fernández-Sáez, Pilar Montesó-Curto

**Affiliations:** 1Internal Medicine Unit, Sant Pau i Santa Tecla Hospital, 43003 Tarragona, Spain; ilga.ruschak@urv.cat; 2Faculty and Department of Nursing, Rovira i Virgili University, 43003 Tarragona, Spain; 3Department of Psychology, Luther College, Decorah, IA 52101, USA; touslo01@luther.edu; 4Rheumatologist at the Santa Maria Hospital, 25198 Lleida, Spain; lrosello@comll.cat; 5Unit of Research Support, Foundation University Institute for Primary Care Research (IDIAP) Jordi Gol, 43500 Tortosa, Spain; caguilar.ebre.ics@gencat.cat (C.A.M.); jfernandez@idiapjgol.info (J.F.-S.); 6Evaluation Unit, Primary Care Direction in Terres de l’Ebre, Institut Català de la Salut (ICS), 43500 Tortosa, Spain; 7Faculty and Department of Nursing, Rovira i Virgili University, 43500 Tortosa, Spain; 8Master in Aging and Health, Department of Medicine, Rovira i Virgili University, 43201 Reus, Spain; 9Primary Care in Institut Català de la Salut (ICS), 43500 Tortosa, Spain

**Keywords:** fibromyalgia, men, opioid, pharmacological-therapy, symptomatology

## Abstract

Fibromyalgia syndrome (FMS) is characterized by generalized chronic musculoskeletal pain, fatigue, and sleep disturbance, as well as cognitive, somatic, and other symptoms. Most people affected by FMS are women, and studies analyzing this condition in men are scarce. In this study, we discuss the physical and psychological symptoms of FMS in men, analyze the possible side effects of pharmacological therapies, and explore the impact of the illness comparing these results between the different classification groups according to sociodemographic variables (marital status, level of education, employment situation and number of people living at home). We used a sequential exploratory mixed method (MM). Qualitative information was obtained from two focus groups (n = 10). Structured questionnaires were administered to 23 men affected by FMS. The mean age of the participants was 51.7 years (SD = 9.64). The most common drugs used were antidepressants and anxiolytics (86.9%), followed by non-steroidal anti-inflammatory drugs (82.6%) and opioids (60.9%). Current level of pain was high (8.2; SD = 1.1), while perceived health and satisfaction with pharmacological treatments were low (4.6; SD = 2.6 and 3.5; SD = 3.2, respectively). The impact of FMS measured using the Fibromyalgia Impact Questionnaire (FIQ) was very high at 88.7 (SD = 8.2). Six categories related with symptoms and side effects of the medication were observed in the qualitative data: (1) main physical symptoms, (2) mood disorders, (3) insomnia and non-restorative sleep, (4) cognitive disturbance, (5) hypersensitivity, and (6) symptoms secondary to opioids. Pain and fatigue were the symptoms most often mentioned by the participants (70% and 80%, respectively). Other important symptoms were anxiety, depression, and memory and sleep disorders. The consumption of opioids causes further unwanted symptoms such as drowsiness and dependence, which makes it difficult for patients to perform basic everyday activities. We believe it is vitally important to continue investigating this symptomatology in order to improve diagnosis and treatment for these patients.

## 1. Introduction

Fibromyalgia syndrome (FMS) is characterized by chronic pain, fatigue, insomnia, cognitive dysfunction, mental health disorders, and other symptoms [1]. As the pathogenesis of FMS is still poorly understood, its diagnosis, based on criteria updated in 2016 by the American College of Rheumatology, is difficult. There are currently no biomarkers of the condition, so clinical evaluation and patient explanations are employed [2].

The prevalence of FMS ranges from 2% to 6% of the world’s population [3,4]. The condition greatly alters an individual’s health-related quality of life. The vast majority of patients are affected, to varying degrees, by disability, social isolation, and stigmatization, while lack of legitimacy regarding diagnosis and uncertainty regarding long-term prognosis also result [5].

The aim of treatment, which is not curative, is to reduce symptoms and afford the patient greater functionality [6]. Given the wide range of signs and symptoms and the low efficacy of standard medical procedures, the numerous interventions differ in their duration, objectives, therapeutic components, and type of professionals. Nevertheless, since FMS is a highly complex condition, most authors conclude that treatment should be holistic, comprehensive, and approached from a multidisciplinary perspective [7].

Another important aspect is the strong predominance of female patients, who represent 80–96% of all FMS patients [3,4]. Since most research conducted to date has focused predominantly on women, understanding of the condition from the male perspective is limited [8]. Eminent researchers in the field, such as Wolfe et al. [4], believe that men with FMS are underdiagnosed, and this may affect the accuracy of statistics relating to symptoms, prevalence, costs, and clinical results. A systematic review of FMS in men and women around the world reported that the prevalence of the condition is similar for both sexes, i.e., roughly 3.98% in women and 2.40% in men [9]. However, the symptoms experienced by men were much less likely to be identified and diagnosed than those of women. The discrepancy between men and women in relation to the prevalence and diagnosis of FMS seems to be related to the social stigma associated with it being considered a “female illness” [8,10].

Studies comparing the symptomatology of women and men with FMS report that men are less sensitive to pain and have fewer tender points [4,11,12,13]. However, when Sánchez et al. [14] explored gender-related differences in the experience of pain, physical activity, and psychological measures with healthy control groups, they reported no significant differences in any variable. Studies on fatigue in FMS by gender have also produced uneven results. For example, Aparicio et al. [15] observed that men reported less fatigue than women. On the other hand, Buskila [11] found that men reported more fatigue than women, while Yunus, Celiker, and Aldag [13] reported no significant differences. Men also manifested more alterations in their sleep patterns [16], as well as greater physical deterioration and worse quality of life [11,17]. Studies of anxiety, stress, and depression by gender also provide results with no significant differences [13]. On the other hand, Buskila and colleagues [11] found that men had a greater range of mental disorders and a poorer quality of life in general [11,15].

Studies focusing entirely on the symptomatology and experiences of men with FMS are scarce. In relation to walking ability, Heredia-Jimenez and Soto-Hermoso [18] reported that men with FMS displayed less speed, a shorter stride, and fewer steps per minute than control groups. Karper [19], for his part, reported positive findings in the maintenance of similar levels of functional capacity in two elderly men with FMS over a four-year period thanks to a program of gentle and continuous walking exercises. However, pain and fatigue are extremely limiting on a daily basis, with periods of calm fluctuating with periods of excruciating difficulty [10,20]. Sexual ability is also impaired since pain is related negatively to sexual satisfaction [21]. Men also suffer a range of cognitive disorders, from occasional problems with concentration to severe and complex memory issues that negatively impact their work and social lives [22]. Men also report a great deal of anxiety, especially about their financial situation, since most are unable to work full time. Men also reported frustration due to a lack of understanding of their symptoms and even in-credulity on the part of healthcare staff, so they often felt neglected by the healthcare system [23]. This perceived absence of a receptive listener suggests that healthcare staff should be provided more education and training in their role in improving the overall health of these patients, whose therapeutic non-compliance will only increase if they believe they are not being listened to and their communication with healthcare providers is ineffective [24]. Paulson and colleagues [23] reported that men had difficulties in expressing their feelings because they were afraid of being described as “weak”, an observation attributed to social rules which dictate that men should not cry or complain as this would result in a loss of masculinity. Sallinen et al. [25] explored the interaction between men and FMS and observed a reconstruction of their participants’ masculinity after diagnosis. It is necessary, therefore, not only to learn how to manage men’s symptoms but also to find coherence in their lives via a new identity that is acceptable both to the individual and the community.

Taking into account the limited evidence provided by clinical manifestations of men with FMS, in this study, we aim to examine the physical and psychological symptoms of FMS in men, analyze the possible side effects of pharmacological therapies, and explore the impact of the illness, comparing the results between different sociodemographic groups (marital status, level of education, employment situation, and number of people living at home).

## 2. Materials and Methods

We used the sequential exploratory mixed method whose central component is qualitative methodology. Qualitative data are collected and analyzed first, followed by quantitative data, which are used primarily to augment qualitative data. Data analysis is connected, and integration occurs in the data interpretation stage and in the discussion [26]. For the qualitative phase, we recruited 10 men from a list provided by the Central Sensitization Syndromes Unit of Santa Maria Hospital in Lleida, Spain. 

The type of sampling used was non-probabilistic or purposive [27]. All men were previously diagnosed with FMS by a rheumatologist using the American College of Rheumatology criteria [2]. Men who agreed to take part in the study were selected. Participants were included if they were over 18 years of age, had been diagnosed with FMS, and lived in Spain. Anyone diagnosed with dementia, schizophrenia, or bipolarity were excluded. 

Subsequently, for the quantitative phase, the selection criteria were the same as the qualitative phase. We could obtain 13 more men (23 in total) from the same list. No participant dropped out of the study. All participants signed an informed consent form, a participant information sheet, and a voice-recording consent and release form.

### 2.1. Qualitative Data Collection

Focus groups were used. Unlike semi-structured interviews, this method uses group interaction as a direct data collection method [28]. Two focus groups, each with sessions lasting roughly two hours and held between May and June 2018, were guided by the following questions and sub-questions. A panel of experts assessed the face and content validity of the questions. This panel was composed of a rheumatologist, three members of his team, and a nurse. Some of the questions were as follows: “What physical and mental symptoms cause you to experience this illness in your body?”“What difficulties do you encounter?” “What causes this difficulty? Fatigue, pain or other symptoms?”“You mentioned that you wake up tired. Don’t you feel better in the morning after sleeping?”“What medicines do you take?” “Do you feel any side effects from taking those medicines?”

The sessions were led by two authors of this study, both of whom are healthcare professionals. The first author is a Ph.D. student and the last is a Ph.D. supervisor, an expert in qualitative analysis. Each session was audio-recorded. The audio files were password-protected and professionally transcribed verbatim by the first and the last authors. Data collection ended when data saturation was achieved.

### 2.2. Qualitative Data Analysis

The transcriptions were examined using content analysis and inductive coding [29]. The first and the last authors (present at the focus groups) read and reviewed the transcriptions; many joint meetings were necessary. To reach a consensus, virtual meetings were held with the rest of the authors, who contributed diverse perspectives and experiences. All opinions were listened to, analyzed, and debated to reach a general consensus. FMS symptoms and the side effects of the pharmacological therapies were coded using Atlas.ti 8 data-management software. The codes were then compared in order to identify similarities and differences and divided into categories and subcategories [29]. All initial categories and subcategories, as well as those that emerged a posteriori, were discussed by the research team as a whole.

### 2.3. Quantitative Data Collection

An ad hoc questionnaire was used between March and May 2018 to obtain the participants’ sociodemographic data. The variables used in the questionnaire were age, marital status, place of residence, nationality, educational level, employment status, and number of people living in their home. The questionnaire also included the patients’ perceived levels of support (from 0 to 10, with a higher value indicating a higher level of support), pain (from 0 to 10, with a higher value indicating a higher level of pain), and health (from 0 to 10, with a higher value indicating a higher level of health), as well as the pharmacological therapies used and the patients’ level of satisfaction with their treatment (from 0 to 10, with a higher value indicating a greater level of satisfaction). No analysis of validation, consistency, etc., was performed and we consider that, due to the type of information required, it was not necessary.

The Fibromyalgia Impact Questionnaire (FIQ) was used to analyze the impact of the illness. This questionnaire is widely used in the healthcare field [30] to assess the status, progression, and prognosis of patients with FMS by measuring aspects of their current health status that are considered to be most affected by the condition. It comprises 10 items, each evaluated on a scale of 0 to 10, and the highest possible score is 100—the higher the score, the greater the impact of the illness. The FIQ asks patients to answer questions on their ability to perform certain tasks in the previous week, such as how many days they felt well, how many days they missed work due to FMS, how they felt in general, and how much their pain or other symptoms made it difficult for them to perform their duties at work. Previous research has shown the FIQ questionnaire to be a valid test of the impact of fibromyalgia [31]. In the present study, Cronbach’s alpha = 0.909.

### 2.4. Quantitative Data Analysis

A descriptive analysis of frequencies and percentages was conducted to reflect the participants’ sociodemographic variables and pharmacological treatments. For continuous variables, a descriptive analysis was performed using mean, standard deviation, maximum, median, and minimum. 

To detect statistically significant differences in the score of the FIQ according to the categorical variables, the non-parametric tests of the Mann–Whitney (two groups) and Kruskal–Wallis (multiple groups) were used. Statistical significance for these tests was set at *p* < 0.05. Non-parametric tests are based on measures of sample position and not on statistical parameters and are therefore less sensitive to outliers.

In the case of normality of variables, the results of parametric and non-parametric tests are similar. In the case of non-normality, the results of the non-parametric tests are more reliable. For “small” samples, these nonparametric tests are more robust. The data were collected and refined using a Microsoft Office Excel spreadsheet. Statistical analysis was conducted using IBM SPSS Statistics v.23.0.

## 3. Results

### 3.1. Qualitative Findings

From the transcriptions of the sessions with the focus groups, we coded six categories and 23 subcategories available in Table 1. The six categories related with symptoms and side effects of the medication were (1) main physical symptoms, (2) mood disorders, (3) insomnia and non-restorative sleep, (4) cognitive disturbance, (5) hypersensitivity, and (6) symptoms secondary to opioids.

#### 3.1.1. Main Physical Symptoms

The participants explained in great detail many of the signs and symptoms caused by FMS but expressed special concern with regard to pain and fatigue. They described their pain as continuous, “a constant struggle” and present “all year round”. Some men also described it as a stabbing pain, “like a heart attack” or “like a sword” (P1). They also described it as generalized and unbearable, to the point of wanting to “hit myself against the wall” (P4, P7). The pain fluctuated in the sense that it was present in different parts of the body, sometimes started in one area and moved to another, and was sometimes present in different areas at the same time.

Fatigue was described as continuous and a daily presence in their lives. Some explained that it appeared early in the morning and did not go away even with rest: “you just wake up tired” (P4). The fatigue was also said to limit their activity, like a battery that is running out, and that it worsened with movement, conditioning simple actions such as “peeling potatoes and having to take breaks to be able to finish the job” (P1). Others said they could not even hold objects in their hands. They had to manage their fatigue well and give priority to certain actions. “(…) I sit down and think, what am I going to do today? My life revolves around seconds because you don’t know what you’re going to find next…” (P8).

#### 3.1.2. Mood Disorders 

The participants described their experiences with anxiety and depression. They attributed their anxiety to a large extent to the uncertainty they experienced before receiving their diagnosis: “(…) the doctors tell you that you’re somaticizing. I suffer from anxiety because I’ve had pain and been taking tests for years” (P3). Failed diagnoses and a lack of empathy displayed by some health professionals as well as by their family and friends have had a negative impact on the mental health of these patients: “People laughed at me and thought I was making it up” (P5). Their constant pain has generated anxiety and led to depression: “Having to put up with so much pain makes you feel depressed” (P3). Depression has also led them to hit rock bottom and to think of suicide: “I’ve wanted to commit suicide, but I haven’t done so for the sake of my daughter” (P10).

#### 3.1.3. Insomnia and Non-restorative Sleep

Most participants do not sleep or rest very well, which they associate above all with their pain and anxiety. Some, for example, explained that they “would wake up in the middle of the night with panic attacks” (P5), while others could not fall asleep because of their “obsessive thoughts” (P3). Lack of sleep also altered their personality because they grow angry more often about things that had never affected them before: “(...) you’re in a bad mood and when you’ve had enough, you get angry with people; they tell you you’ve changed, that you get upset at the slightest things” (P6). With regard to pharmacotherapy, the participants explained that they take sleeping pills to get to sleep: “I couldn’t get any rest until they prescribed me some strong sleeping pills” (P9). They also take anxiolytics with hypnotic effects: “I take diazepam (Valium) to get to sleep; I couldn’t get to sleep without it” (P1).

#### 3.1.4. Cognitive Disturbance

Another symptom mentioned by participants at the focus group sessions was memory impairment. They reported having a poorer memory and feeling more forgetful, especially as the illness progressed: “I forget about things a lot” (P8). Some participants at the sessions could not remember what they wanted to say when it was their turn to speak. Others said they had difficulty expressing themselves verbally and that this may have been caused by their mixing and consuming a range of different drugs.

#### 3.1.5. Hypersensitivity

All participants reported having some form of hypersensitivity. With regard to auditory hypersensitivity, for example, one participant reported that his eardrums “pierced” (P4) whenever he heard low-pitched noises, while another explained that he appreciated silence much more than he had before. Others described their hypersensitivity in more temperature-related terms. Some were affected by changes in temperature, while others could not stand the summer heat because it exacerbated their fatigue. Some complained that air conditioning intensified their pain or that they had to keep out of the sun because they burned easily. Others said they could not speak in cold weather because it made their tongues feel all prickly. They also reported being hypersensitive to chemical odors such as colognes and lacquers, which made them feel nauseous or dizzy or gave them headaches: “I avoid the perfume aisles in supermarkets” (P5).

#### 3.1.6. Symptoms Secondary to Opioids

Many participants reported that although they felt their drugs were effective, they were unsure whether to take them because they led to unwanted side effects such as dependence, impaired daily functionality, or interaction with other substances such as alcohol. Although highly effective, powerful painkillers such as opioids led to feelings of “drowsiness” or “extreme relaxation” that made it impossible for them to lead their normal daily lives: “Strong medicines make you feel sleepy and you can’t lead a normal life or do the things you need to do for yourself” (P9); “I’ve even picked the girls up from school when I’ve been drugged up with morphine.” (P1). The same participant recognized that morphine only served to hide his pain: “It took me two years to get off morphine; I didn’t want to end up taking methadone, it just didn’t make any sense” (P1). Another major side effect, therefore, is dependency. Some men were very sincere: “I’m hooked on opioids and I can’t stop taking them” (P7). One participant explained that he always carried some “candy” with him. Others reported they had stopped taking medication for a while but ended up re-taking it because they could not cope with the pain.

### 3.2. Quantitative Findings

The sociodemographic data of the 23 participants are shown in Table 2. The mean age of the participants was 51.7 years (SD = 9.64 years). Of these participants, 60.9% were married or with a partner, while 21.7% were separated or divorced; 26.1% lived alone, while 65.2% lived in a home comprising two to four people; only 13% of the participants worked, while 26.1% were retired; and 21.7% were unemployed, while 17.4% were off work with permanent disability. The average score for the perceived level of support from the family environment was 7 (SD = 2.5), while the average score for the perceived level of support from outsiders such as friends was 3 (SD = 1.3).

Table 3 shows the pharmacological therapies used. The participants were found to consume a large number of drugs to alleviate their symptoms. The drugs par excellence were antidepressants and anxiolytics (86.9%), followed by analgesics known as NSAIDs (non-steroidal anti-inflammatory drugs) (82.6%). Strong analgesics such as opioids accounted for 60.9% of the drugs taken by the participants, while Lyrica, for neuropathic pain, accounted for 52.2% and infiltrations accounted for 43.5%. Other drugs accounted for 12.9%. The participants’ average level of satisfaction with their pharmacological treatments was 3.5 (SD = 3.2), while their perceived average level of pain at the time they completed the questionnaire was 8.2 (SD = 1.1) and their perceived average level of health was 4.6 (SD = 2.6).

The results of the FIQ questionnaire (Table 4) show that the average score for the daily life activities ranged from 0 to 3. The activities most affected were preparing food (M = 2.5; SD = 1.0), walking several hundred meters (M = 2.5; SD = 1.2), and driving (M = 2.5; SD = 1.2). With regard to the participants’ perceptions of the symptoms they had experienced in the previous week, the scores, which ranged from 0 to 10 (with higher scores indicating a worse perception), were as follows (from highest to lowest): feeling of tiredness (M = 8.9; SD = 1.2); feeling of tension, nervousness, and anxiety (M = 8.5; SD = 1.3); feeling of stiffness (M = 8.4; SD = 1.5); feeling of pain (M = 8.3; SD = 1.6); difficulty waking up in the morning (M = 8.3; SD = 1.9); and feeling sad or depressed (M = 8; SD = 2.4).

On average, the ability of the participants to work was scored as 7.4 (SD = 2.9). Due to their condition, they lost 3.8 days a week for work or tasks (SD = 2.6) and only felt well 1 day (SD = 1.8). The impact of the disease for all participants was very high, while the mean for the FIQ as a whole was M = 88.7 (SD = 8.2).

Table 5 shows the values of total FIQ according to participants’ sociodemographic variables such as marital status, academic level, employment situation, and number of people living at their home. The table shows that the impact of the illness is slightly, though not statistically, greater in married men or men with a partner (M = 91.7, SD = 5.8) as compared to unmarried men (M = 88.3, SD = 8.6); those who have completed secondary or university education (M = 89.4, SD = 7.9) as compared to those who have completed primary education (M = 86.8, SD = 9.8); the unemployed (M = 95.2, SD = 3.6) as compared to those who were employed (M = 85.3, SD = 10.6), active with work disability (M = 89.8, SD = 4.1), had permanent work disability (M = 83.0, SD = 1.4), and retirees/pensioners (M = 87.8, SD = 12.3); and those living with more than two people (M = 90.0, SD = 6.8) as compared to those living alone (M = 85.0; SD = 11.3).

Individually, the greatest impact was reported by participant number 4 (55 years old; FIQ = 93.5), who was divorced, perceived a poor level of family support (4/10), had completed secondary education, was unemployed due to a disability, and consumed opioids. The least impact was reported by participant number 9 (50 years old; FIQ = 74.2). Although the profile of this participant was similar to participant number 4′s, he was younger and married, had better perceived family support (10/10), and did not consume opioids.

The results obtained through the mixed method suggest that the most common physical symptoms of our participants were pain and fatigue, which caused difficulties in activities of daily living (ADL). Anxiety and depression were also very present. Despite consuming mainly anxiolytics, antidepressants, and strong analgesics, the men were not satisfied with pharmacological treatment. In summary, there is a high impact of the disease on participants of the study.

## 4. Discussion

Our quantitative results depicted participants as middle-aged men (51.7 years), as are the vast majority of studies in men with FMS, which reveal young participants with mean ages of 37 [21], 45 [18], 47 [22,25], 48 [23], and 52 years [20]. The only study that presented older participants was that of Karper [19], but he presented only two men, who were 61 and 69 years old. Most of the study participants were married or had a partner, as in the articles by Sallinen et al. [22,25], Kueny et al. [20], and Muraleetharan et al. [24]. In terms of work activity, less than half of our participants had a job, unlike the studies by Sallinen et al. [22,25] and Paulson et al. [23], where the majority were working or retired.

Regarding pharmacological treatment, our results agree with Rivera et al. [32], who state that the drugs most commonly used by patients with FMS are NSAIDs, benzodiazepines (anxiolytics/hypnotics), some anticonvulsants, and major opioids. In our study, a high use of NSAIDs was detected (82.6%). These are used very frequently in patients with FMS even though there are insufficient studies to recommend their use for pain management [32,33]. Their use should be rationed due to potential gastrointestinal, renal, and cardiovascular adverse effects [32]. The high use of opioids, which accounted for 60.9%, was also surprising. The use of opioids in patients with FMS is not usually effective because these patients have altered endogenous opioid activity, with low availability of opioid receptors [33]. The only opioid that has been shown to be effective in patients with fibromyalgia is tramadol (weak opioid) alone or in combination [32,33,34]. This probably reflects the severity of pain and associated disability in these patients, as well as the general limitation of all available pharmacological treatments for pain [35]. The use of opioids for the treatment of chronic nononcologic pain has increased dramatically in recent decades [32]. Opioid abuse for the treatment of chronic pain is a problem that is not limited to the USA, but also seems to affect European countries such as the one in our study (Spain) [35]. 

In summary, patients with FMS in Spain are overtreated with a combination of pharmacological therapies that lack adequate support from clinical practice guidelines, and some drugs that have not been shown to be effective and are harmful in the long term, such as major opioids, benzodiazepines, and NSAIDs, should be dispensed with [32,35]. This results in dissatisfaction on the part of the patients [35], which is also present in our study, with the participants’ mean level of satisfaction with their pharmacological treatment being 3.5 (SD = 3.2).

Qualitative results, on the other hand, revealed that pain and fatigue were the clinical manifestations most often mentioned by the participants [20]. Pain, which was described here as chronic and generalized, is reported to be the main symptom of fibromyalgia [1]. The participants reported their painful experiences in great detail, defining pain as continuous, present throughout the year, and fluctuating in that it can begin in one part of the body and move to another. The intensity of the pain may also fluctuate during the day [20,23]. Being highly unpredictable, the pain causes much uncertainty, which makes it difficult to plan ahead. The patients have to live with intense pain on a daily basis, to the point that they sometimes want to “hit themselves against the wall”. One participant even complained that a life free from pain should be a basic right for all. This intense pain notwithstanding, some days are calmer. Like participants in the study by Sallinen and Mengshoel [1], participants in our study explained that, although some days were better than others, they could not remember what it was like to live a whole day free from pain. 

Our findings suggest that fatigue is a symptom that strongly limits every aspect of patients’ lives. It fluctuates throughout the day and across all four seasons. Fatigue is compared to a battery gradually running out with every passing hour. Simple activities such as cooking or using a screwdriver are big problems because they have to take constant rests when performing any kind of task. They also mentioned that their fatigue is cumulative and does not subside completely with rest [20]. It is impossible to wake up without fatigue, and since fatigue is present first thing in the morning, they find it difficult just to get out of bed. A task as simple as making the bed is also beyond them, and some of them choose never to make it.

Another important symptom in patients with FMS is sleep disorder. As in a study of women affected by FMS and healthy control groups [36], participants in our study reported that poor sleep quality or insomnia directly contributes to greater fatigue. Pain, fatigue, and insomnia are thus closely linked. The lack of quality sleep leads to morning fatigue, which leads to less mobility, which in turn causes more pain. Simple physical tasks become a brick wall. As in the study by Sallinen and Mengshoel [1], the present study data suggest that the altered psychological state of men is a direct consequence of their chronic pain and fatigue, i.e., the deterioration in their health irritates them so much that they suffer from mental health problems. Participants reported feeling not only physical but also mental fatigue that leads to anxiety, depression, negative thoughts, and suicidal ideas [36]. They also attributed this effect on the psychological state to negative experiences, comorbidity with other illnesses, financial problems, failed diagnoses, suffering from a little-known illness, suffering from a “women’s illness”, waiting for a recovery that never arrives, a lack of empathetic listeners, and generally losing their previous life [10,20].

Participants in the present study also expressed special concern for memory impairment and cognitive dysfunction, also known as “fibrofog” [37]. Our data showed they often forget things and feel more absent-minded and that they sometimes dry up in the middle of a conversation. Kravitz and Katz in their review [38] reported the presence of cognitive deficits, which is a common, distressing, and disabling symptom among such patients. Fibromyalgia patients also display a poorer working memory compared to healthy individuals, while their short- and long-term memories also appear to be impaired [37].

FMS is grouped with other diseases as one of the Central Sensitization Syndromes [5]. These refer to individuals who are more sensitive to the sensory information they receive and are thus more prone to certain hypersensitivities [39]. Participants in the present study described symptoms such as auditory hypersensitivity to certain noises, thermal hypersensitivity to high and low temperatures, and hypersensitivity to odors from chemical products such as bleach, colognes, and ointments [39,40,41,42]. Auditory hypersensitivity in these patients has not yet been analyzed, however. We believe that these symptoms may be closely related to the irritability they experience due to FMS. With regard to thermal sensitivity, a recent study showed that people with chronic pain often report that their symptoms are aggravated by weather conditions. Participants in the study were seen to be more sensitive to hot and cold ambient temperatures than pain-free control subjects [38]. When preparing strategies to help such patients cope with hot or cold environments, for example, it would be interesting to remember that temperature can affect people with FMS. Some participants mentioned discomfort and headaches secondary to certain smells and chemical products such as colognes, bleaches, and air fresheners. However, research into hypersensitivity to certain odors or chemicals in patients with FMS is scarce. Sayılır and Çullu [41] reported that the olfactory function of patients with FMS is different from that of healthy control subjects. Using cranial MRI, they associated a low-volume olfactory bulb with impaired olfactory functions such as dentification, threshold, and discrimination with regard to certain odors. The above authors suggest that a smaller olfactory bulb may be caused by FMS-related changes to the CNS in the neural structures of affected individuals. 

Finally, we observed that participants in the present study suffer from symptoms secondary to the drugs they take, especially opioids. The use of opioids for pain management in FMS is not recommended by any current clinical guidelines [32]. The men in our study reported being dependent on such substances. According to Fitzcharles and colleagues [43], people with FMS who took opioids showed signs of worse functional and cognitive status. In agreement with the participants in the above study, participants in the present study also reported having their daily functionality impaired, feeling drugged, and lacking energy. They also associated opioid use with reduced deep sleep and increased light sleep. Although opioids are believed to have sedative effects and are often prescribed to reduce pain and promote sleep onset, other results suggest that they may actually have the opposite effect [44]. Long-term opioid use is also associated with being unemployed and having a history of abuse with other substances, such as alcohol and illicit drugs [43].

In short, men with FMS are people who suffer from a chronic disease at a very early age, suffering numerous physical and mental symptoms that make it impossible to work or carry out basic life activities. Their pharmacological treatment is unsatisfactory, and they take drugs that are not recommended by clinical guidelines because they are ineffective or cause serious side effects. These findings show an important deficiency in the lack of research in relation to FMS, as well as a lack of male participants, lack of knowledge of FMS symptomatology in relation to men, and lack of effective and safe treatment.

## 5. Conclusions

This study has limitations. The sample size is small. This reduces the variability in the sample composition, limits generalization, and reduces statistical power for quantitative analyses. A greater number of participants may have increased sample diversity and have added new perspectives to the discussion. It should be noted, however, that this study has a larger and more diverse sample than many other studies that are currently available and it does capitalize on a mixed-methods approach. Another important limitation is that the data are entirely patient-reported, which may result in under- or over-reporting. 

Despite the limits of this study, our results show that it is extremely important to continue investigating this subject to better understand the sociodemographic and clinical characteristics and perceptions of these patients. This study may be particularly useful in the public or community healthcare fields to help boost the visibility and recognition of male FMS patients, and in private healthcare, it could be an effective way to organize the future service of the providers.

## Figures and Tables

**Table 1 ijerph-19-01724-t001:** Categories and quotes on the symptomatology of men with FMS.

CATEGORIES	QUOTES
**1. Main physical symptoms**	
**PAIN**	**Continuous pain** *“I have pain all year long.” (P1, P2, P10)* *“It’s a struggle with continuous pain.” (P2)* **Stabbing pain** *“It’s like a heart attack, the chest pain.” (P1, P3, P4, P8)* *“The pain is like a sword: it enters the front of your chest and comes out at the back.” (P1)* **Generalized pain** *“Everything hurts and it makes me want to hit myself against the wall.” (P4, P7)* *“When I raise my arm to change a light bulb, it hurts all day.” (P8)* **Unbearable pain** *“Even my soul aches.” (P10)* *“Everything except my eyelashes hurts.” (P10)* **Fluctuating pain** *“Suddenly the pain starts to paralyze one part of my body, and then another.” (P4)* *“There are times when one arm hurts, then the other, and then both at the same time.” (P2)*
**FATIGUE**	**Constant fatigue** *“Fatigue is constant throughout the year.” (P5, P10)* *“I couldn’t hold any object in my hands.” (P4, P9)* **Morning fatigue** *“You have a hard time getting out of bed.” (P6, P7, P10)* *“You wake up tired.” (P4)* **Limiting fatigue** *“Tiredness is like your batteries have gone dead.” (P5)* *“You grab a potato, then you have to wait to recover for a while before carrying on.” (P1)* **Managing fatigue** *“I wake up, then I sit down and think, what am I going to do today? You don’t know whether fatigue will hit you in a minute; my life revolves around seconds because you don’t know what you’re going to find next... “ (P8)*
**2. Mood disorders**	
**ANXIETY**	**Uncertainty/Tiredness** *“I suffer from anxiety because of the multiple diagnostic tests and constant pain.” (P3, P10)* *“I suffer from anxiety because I’ve had pain and been taking tests for years. At first you think it will go away but after some failed and serious diagnoses you end up suffering a great deal of anxiety.” (P3)* **Lack of empathy** *“You go to a specialist and he tells you you’re anxious, that you’re somatizing.” (P3)* *“They declared me permanently disabled. I keep meeting people who were friends and who tell me how lucky I am! And then I say, would you like to swap your job for my pain and disability?” (P4)* *“Few people understand this disease.” (P1, P6, P9)* *“People laughed at me and thought I was making it up.” (P5)*
**DEPRESSION**	**Depression secondary to pain** *“The pain made me hit rock bottom, so the doctor referred me to a psychiatrist.” (P9)* *“Having to put up with so much pain makes you feel depressed.” (P3) * **Suicidal thoughts** *“I still get negative thoughts and ideas.” (P3)* *“I’ve wanted to commit suicide, but I haven’t done so for the sake of my daughter.” (P10)*
**3. Insomnia and non-restorative sleep**	
**INSOMNIA**	**Insomnia secondary to pain** *“The pain stops me from going to sleep.” (P6)* **Insomnia secondary to anxiety** *“I can’t sleep because of the anxiety.” (P5, P9)* *“I couldn’t sleep; I kept waking up with anxiety, panic attacks and nightmares.” (P5)* *“I have obsessive thoughts and nightmares, and sometimes I pee in bed.” (P3)* **Irascibility** *“(…) you’re in a bad mood and when you’ve had enough you get angry with people; they tell you you’ve changed, that you get upset at the slightest things.” (P6)* *“Every day I tolerate people less and less; I can’t concentrate when several people are speaking at the same time” (P10)* **Pharmacotherapy** *“I took a lot of antidepressants and anxiolytics because I had insomnia, but in the morning, I felt all drugged up.” (P5)* *“Until they prescribed me some strong sleeping pills, I couldn’t get any rest, I couldn’t sleep.” (P9)* *“I take diazepam (Valium®) to get to sleep; I couldn’t get to sleep without it” (P1).*
**4. Cognitive disturbance**	
**MEMORY**	**Memory impairment** *“I forget about things a lot.” (P8)* *“I take a lot of medication, but I can’t even remember what half of it is for.” (P4)* *“You have something to say and then when you’re going to say it, you forget what you were going to say.” (P1)* *“I have trouble expressing myself.” (P4)*
**5. Hypersensitivity**	
**HYPERSENSITIVITY**	**Auditory hypersensitivity** *“If someone makes a low-pitched noise, it’s as if they were piercing my eardrums.” (P4)* *“I like silence. I appreciate it more now than before.” (P10)* **Thermal hypersensitivity** *“When I used to do water aerobics, the water was warm for my friends but not for me. When I go underwater, it smarts.” (P4)* *“Changes in temperature affect me. I can’t stand the heat in summer. I can’t even talk about the cold in winter; my tongue gets all prickly and inflamed.” (P1)* *“The air conditioning makes my pain feel worse.” (P3, P4)* *“I wear T-shirts and put on sunblock to protect myself. I get burned a lot in the sun.” (P3)* **Chemical hypersensitivity** *“I can’t breathe and my head hurts when I’m near perfumes, colognes, lacquers, creams and bleaches.” (P1)* *“I avoid the perfume aisles in supermarkets.” (P5)*
**6. Symptoms secondary to opioids**	
**THE EFFECTS OF OPIOIDS**	**Drowsiness/drug overdose** *“They prescribed me medicines that got me high.” (P4)* *“I’ve even picked the girls up from school when I’ve been drugged up with morphine.” (P1)* *“If you drink any alcohol with these drugs, then you’ve really messed up.” (P6)* *“In the end you can’t even remember your own name.” (P8)* *“I’ve taken a lot of strong medicines; they make you feel sleepy and you can’t lead a normal life or do the things you need to do for yourself.” (P9) * **Dependence** *“The morphine doesn’t do anything; it just takes away the pain. It took me two years to get off it; I didn’t want to end up taking methadone, it just didn’t make any sense.” (P1)* *“I’m hooked on opioids; I’ve tried to stop taking them but after three days I really needed them so started taking them again. I always carry some ‘candy’ (morphine) with me.” (P7)* *” I’ve learned to take morphine and then it’s done.” (P7)* *“I’ve gone through stages when I’ve not taken anything but then you’ve got to rush out and grab the pills.” (P8)*

**Table 2 ijerph-19-01724-t002:** Characteristics of the participants.

CHARACTERISTICS	*n* = 23	%
**Marital status**		
Single	3	13.1
Married or men with a partner	14	60.9
Divorced or separated	5	21.7
Widower	1	4.3
**Place of residence**		
Lleida	15	65.2
County of Lleida	8	34.8
**Nationality**		
Spanish	22	95.7
Romanian	1	4.3
**Level of education**		
Primary education	6	26.1
Secondary education	15	65.2
University education	2	8.7
**Employment situation**		
Employed	3	13.1
Unemployed	5	21.7
Active with work disability	5	21.7
Permanent work disability	4	17.4
Retiree or pensioner	6	26.1
**Number of people living at home**		
One	6	26.1
Between two and four	15	65.2
More than 4	2	8.7
	**M**	**SD**
**Age**	51.7	9.6
**Level of support received**		
Family environment	7.0	2.5
Parents	6.6	3.6
Children	5.6	3.2
Partner	4.9	4.3
Friends	3.0	1.3

**Table 3 ijerph-19-01724-t003:** Pharmacological treatments.

	n	%
Pharmacological Therapies Received To Date		
Antidepressants	20	86.9
Anxiolytics	20	86.9
NSAIDs	19	82.6
Opioids	14	60.9
Lyrica (antiepileptic/neuropathic pain)	12	52.2
Infiltrations	10	43.5
Others	3	12.9
Asthma inhalers	2	8.6
Antipsychotics	1	4.3
Don’t know/no comment	2	8.6
	**M (0–10)**	**SD**
Current pain level	8.2	1.1
Perceived health level	4.6	2.6
Satisfaction with pharmacological treatments	3.5	3.2

Note: NSAIDs = non-steroidal anti-inflammatory drugs.

**Table 4 ijerph-19-01724-t004:** FIQ items (daily life activities).

	**M**	**SD**
Are you able to go shopping?	2.5	0.9
Are you able to do the washing in the washing machine?	2.4	1.0
Are you able to prepare the food?	2.5	1.0
Are you able to wash the dishes by hand?	2.4	1.1
Are you able to use the vacuum cleaner?	2.4	1.0
Are you able to make the bed?	2.5	1.0
Are you able to walk several hundred meters?	2.5	1.2
Are you able to visit friends or relatives?	2.3	1.2
Are you able to do the gardening?	1.7	1.1
Are you able to drive a car?	2.5	1.2
Are you able to climb the stairs?	2.3	1.2
	**Mean (0–10)**	**SD**
How tired have you felt?	8.9	1.2
To what extent have you felt tense, nervous or anxious?	8.5	1.3
How stiff have you felt?	8.4	1.5
To what extent have you felt pain?	8.3	1.6
How did you feel when you got up this morning?	8.3	1.9
To what extent have you felt sad or depressed?	8.0	2.4
How much did the pain affect your ability to work?	7.4	2.9
How many days’ work or days for doing things around the house did you miss last week because of your Fibromyalgia?	3.8	2.6
Of the 7 days in the week, how many did you feel well?	1.0	1.8
**Total FIQ**	**88.7**	**8.2**

SD: standard deviation.

**Table 5 ijerph-19-01724-t005:** Total FIQ values according to sociodemographic variables.

	n	Minimum	Median	Maximum	* ^a^ * *p*
**Marital status**					
Married or men with a partner	3	85	95	95	0.404
Single, divorced or separated, widower	20	74	87	104	
**Level of education**					
Primary education	6	74	85	101	0.516
Secondary or university education	17	77	89	104	
**Employment situation**					
In employment	3	74	87	95	0.154
Unemployed	5	91	95	101	
Active with work disability	5	85	89	95	
Permanent work disability	4	81	83.5	84	
Retiree or pensioner	6	77	83.5	104	
**Number of people living at home**					
One	6	74	81	102	0.256
More than 2	17	80	89	104	

a Non-parametric Mann–Whitney test and non-parametric Kruskal–Wallis test.

## Data Availability

The data underlying the findings described in this manuscript are available to interested researchers upon request.

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
