# Peer review of "Symptomatology of Fibromyalgia Syndrome in Men: A Mixed-Method Pilot Study"

_ijerph, 2022, doi:10.3390/ijerph19031724_

Round 1

Reviewer 1 Report

Thank you for the opportunity to review this interesting manuscript. I have a few minor comments for the authors to consider:

  • Abstract:
    • Line 31: Please clarify the “Six categories” mentioned. Were they the categories of symptoms or impact or side effect or something else?
    • Line 33: The authors mentioned that pain and fatigue were the most often mentioned by the participants. Please include the actual results in a bracket, e.g. the most often (x % or n= frequency) mentioned by the participants.
  • Introduction:
    • Lines 94-103: Since the focus of this manuscript is “physical and mental symptoms of men with FMS, side effects of pharmacological therapies, and impact of FMS”, the authors may delete (or rephrase) the sentences around communication with health professionals.
  • Methods:
    • Line 113: Quantitative data do not necessary “improve” qualitative data. Please revise the word “improve” to “complement” qualitative data in order to answer the research questions.
    • Lines 115-121 described the sampling method and should therefore be a new paragraph on its own.
    • Line 122: Please clarify the sampling method and selection criteria for the quantitative phase. If they are the same as the qualitative phase, please state so. If not, please provide details to support the statement about this cohort being “more representative”, which is unlikely to be achieved through the use of a non-random sampling method (e.g. purposive sampling) .
    • Line 130: Please provide details on how the questions were validated. For instance, did a panel of expert assess the face validity and content validity of the questions? Did the authors assess the test-retest reliability and/or internal consistency of the items? If the questions were not validated, the authors need to mention this as a limitation of the study because misclassification bias might have occurred and could affect the findings.
    • Line 153: Please clarify how consensus was achieved if there had been disagreement between members of the research team.
    • Section 2.3:
      • First paragraph. Please provide details on how the questions were validated for its face validity, content validity, test-retest reliability and/or internal consistency etc?
      • Second paragraph: Please quote the validity and reliability score of the Fibromyalgia Impact Questionnaire, e.g. Cronbach’s alpha, etc.
    • Lines 179-180: Please justify the rationale of using the non-parametric tests. Were the outcome variables skewed (i.e. did not have a normal distribution)? If so, please state so, and include the assessments the authors used to assess normality of the continuous outcome variables. In addition, the authors need to add another research question (or research hypothesis) to the Introduction section as an additional aim of this study, in order to support the need of performing the comparison tests.
  • Results:
    • Line 188: Please clarify that there were the 6 categories of symptomology.
    • Lines 301-304: The authors could have described the main results instead of repeated the numbers in the table. For example, the authors could have mentioned that: “On average, the ability of the participants to work was scored as 7.4 (SD=2.9). Due to their condition, they lost 3.8 days in a week for work or task (SD=2.6) and only felt well 1 day (SD=1.8).
    • Table 5: The authors can delete the columns of “M” and “SD” if the values were skewed.
    • The groups results need to be stated in the abstract if the authors revised the aims to include group comparisons.
    • Since this is a mixed method study, it would be good to add a summary paragraph to the end of the results section to link the qualitative and quantitative findings.
  • Discussions:
    • Please delete the first paragraph because there is no need to repeat the results in the discussion section. Instead, please focus on discussing (e.g. comparing the similarities or discrepancies) of the findings against the published literature.
    • Line 344. There is no need to cite a reference when reporting your own findings.
    • Please include a paragraph on the ‘strengths and limitations’ of the present study. For instance, this study has attempted to answer the research gap of the limited evidence, but the authors need to address the limitations of their study design (cross-sectional study, which can affect the investigation around temporal effect), recruitment method (non-random sampling technique, which can affect the representativeness of the findings), validity and reliability of the measurement tool (which can affect measurement bias), etc.
    • The authors could add suggestions for future directions or outline the impact of the current findings to the end of the Discussion section (last paragraph).
  • Conclusion:
    • Lines 436-451 onwards: Please delete the sentences. Conclusion needs to be succinctly presented.

Author Response

Response to Reviewer 1 Comments

Abstract:

Point 1: Line 31: Please clarify the “Six categories” mentioned. Were they the categories of symptoms or impact or side effect or something else?

Response 1: Line 32

Point 2: Line 33: The authors mentioned that pain and fatigue were the most often mentioned by the participants. Please include the actual results in a bracket, e.g. the most often (x % or n= frequency) mentioned by the participants.

Response 2: Line 35-36

Introduction:

Point 3: Lines 94-103: Since the focus of this manuscript is “physical and mental symptoms of men with FMS, side effects of pharmacological therapies, and impact of FMS”, the authors may delete (or rephrase) the sentences around communication with health professionals.

Response 3: Thank you, we rephrased. Lines 97-99.

Methods:

Point 4: Line 113: Quantitative data do not necessary “improve” qualitative data. Please revise the word “improve” to “complement” qualitative data in order to answer the research questions.

Response 4: Thank you in  (Muraleetharan et al, 2018), the word they refered is not improve but augment. Line 120.

Point 5: Lines 115-121 described the sampling method and should therefore be a new paragraph on its own.

Response 5: Lines 129-30.

Point 6: Line 122: Please clarify the sampling method and selection criteria for the quantitative phase. If they are the same as the qualitative phase, please state so. If not, please provide details to support the statement about this cohort being “more representative”, which is unlikely to be achieved through the use of a non-random sampling method (e.g. purposive sampling).

Response 6: Line 129.

Point 7: Line 130: Please provide details on how the questions were validated. For instance, did a panel of expert assess the face validity and content validity of the questions? Did the authors assess the test-retest reliability and/or internal consistency of the items? If the questions were not validated, the authors need to mention this as a limitation of the study because misclassification bias might have occurred and could affect the findings.

Response 7: Lines 137-139.

Point 8: Line 153: Please clarify how consensus was achieved if there had been disagreement between members of the research team.

Response 8: Lines 158-162.

Section 2,3

Point 9: First paragraph. Please provide details on how the questions were validated for its face validity, content validity, test-retest reliability and/or internal consistency etc?

Response 9: Lines 176-78.

Point 10: Second paragraph: Please quote the validity and reliability score of the Fibromyalgia Impact Questionnaire, e.g. Cronbach’s alpha, etc.

Response 10: Lines 188-90.

Point 11: Lines 179-180: Please justify the rationale of using the non-parametric tests. Were the outcome variables skewed (i.e. did not have a normal distribution)? If so, please state so, and include the assessments the authors used to assess normality of the continuous outcome variables. In addition, the authors need to add another research question (or research hypothesis) to the Introduction section as an additional aim of this study, in order to support the need of performing the comparison tests.

Response 11: Lines 199-203.

Results

Point 12: Line 188: Please clarify that there were the 6 categories of symptomology.

Response 12: Lines 209-210.

Point 13: Lines 301-304: The authors could have described the main results instead of repeated the numbers in the table. For example, the authors could have mentioned that: “On average, the ability of the participants to work was scored as 7.4 (SD=2.9). Due to their condition, they lost 3.8 days in a week for work or task (SD=2.6) and only felt well 1 day (SD=1.8).

Response 13: You are absolutely right and the text of the article is changed to the text you propose. Lines 325-27.

Point 14: Table 5: The authors can delete the columns of “M” and “SD” if the values were skewed.

Response 14: In truth M and DS are there to give a view of the sample statisticians in each group. The purpose is simply to show and not to compare. But if these data might confuse the reader, we have determined to remove them from the table. Line 356.

Point 15: The groups results need to be stated in the abstract if the authors revised the aims to include group comparisons.

Response 15: An explanatory sentence in the abstract has been added. Lines 22-24

Point 16: Since this is a mixed method study, it would be good to add a summary paragraph to the end of the results section to link the qualitative and quantitative findings.

Response 16: Lines 350-55.

Discussions

Point 17: Please delete the first paragraph because there is no need to repeat the results in the discussion section. Instead, please focus on discussing (e.g. comparing the similarities or discrepancies) of the findings against the published literature.

Response 17: Lines 360-91.

Point 18: Line 344. There is no need to cite a reference when reporting your own findings.

Response 18: Deleted.

Point 19: Please include a paragraph on the ‘strengths and limitations’ of the present study. For instance, this study has attempted to answer the research gap of the limited evidence, but the authors need to address the limitations of their study design (cross-sectional study, which can affect the investigation around temporal effect), recruitment method (non-random sampling technique, which can affect the representativeness of the findings), validity and reliability of the measurement tool (which can affect measurement bias), etc.

Response 19: In limitations and Conclusions. Line 478.  

Point 20: The authors could add suggestions for future directions or outline the impact of the current findings to the end of the Discussion section (last paragraph).

Response 20: Lines 471-77.

Conclusion

Point 21:  Lines 436-451 onwards: Please delete the sentences. Conclusion needs to be succinctly presented.

Response 21: Lines  478-91.

We have revised the English language and style by one member of the team who is native.

And we changed the data when we did the focus group as we saw we made a mistake. They were holdedin May and June not in January and February.

Reviewer 2 Report

I woukd like to add some comments/questions?

  • In methodology section: If Participants over 18 years of age were included, "anyone under 18.." does not need to be included as an exclusion criteria. (page 3, line 120)
  • Was this study approved by an Ethics Committe? Please, include it or justify.

Author Response

Response to Reviewer 2 Comments

Point 1: In methodology section: If Participants over 18 years of age were included, "anyone under 18.." does not need to be included as an exclusion criteria. (page 3, line 120)

Response 1: Thank you, deleted. Line 127.

Point 2: Was this study approved by an Ethics Committe? Please, include it or justify.

Response 2: It is in the Institutional Review Board Statement section. Line 501.

We have revised the English language and style by one member of the team who is native.

And we changed the data when we did the focus group as we saw we made a mistake. They were holded in May and June not in January and February. Line 136.

Reviewer 3 Report

This study evaluated symptoms and most frequent  drug intake in men affected with Fibromyalgia (FMS), using structured questionnaires.

The topic is of great interest as the vast majority of the studies on FMS are on women and the data on men are very scarce. However, the number of examined subjects in this study is relatively small, and also the entire sample is from one country only, which may impact on the typical drug prescription for the condition. Therefore the data, though interesting, appear to be preliminary:

-I think that this concept should be reflected in the title, by adding, e.g., Preliminary data, or Pilot Study, at the end.

-A surprising finding regards the high level of consumption of opioids in the examined men, although, as the authors correctly underline, opioids are not recommended for FMS. Why was the prescription so high ? I think it is very important for the authors to comment specifically on this issue, eventually compare the current prescription of opioids in the female population in their country as well as in other countries. Is this prescription due to a lack of adequate formation/information on FMS among treating physicians ? Or does it correspond to the presence of other pain comorbidities in the examined patients, which may benefit from this treatment ? FMS is, indeed, quite comorbid with several other pain conditions, both at visceral and somatic level. More info should be provided about specific pain comorbidities in the evaluated patients

-Also the use of NSAIDs seems particularly high : here again, they are not recommended specifically for FMS, unless an inflammatory comorbidity is present, in which case there is an indirect effect of the NSAIDs onto the FMS pain presumably because of a reduction of the noxious load on the Central Nervous System from the peripheral inflammatory areas.

This high use should also be commented by the authors in the light of the available data from the literature

Author Response

Response to Reviewer 3 Comments

Point 1: The topic is of great interest as the vast majority of the studies on FMS are on women and the data on men are very scarce. However, the number of examined subjects in this study is relatively small, and also the entire sample is from one country only, which may impact on the typical drug prescription for the condition. Therefore the data, though interesting, appear to be preliminary:

-I think that this concept should be reflected in the title, by adding, e.g., Preliminary data, or Pilot Study, at the end.

-We agree. We added Pilot in the title. Line 3.

-A surprising finding regards the high level of consumption of opioids in the examined men, although, as the authors correctly underline, opioids are not recommended for FMS. Why was the prescription so high? I think it is very important for the authors to comment specifically on this issue, eventually compare the current prescription of opioids in the female population in their country as well as in other countries. Is this prescription due to a lack of adequate formation/information on FMS among treating physicians ? Or does it correspond to the presence of other pain comorbidities in the examined patients, which may benefit from this treatment? FMS is, indeed, quite comorbid with several other pain conditions, both at visceral and somatic level. More info should be provided about specific pain comorbidities in the evaluated patients

Response 1: Lines 375-384.

Point 2: -Also the use of NSAIDs seems particularly high: here again, they are not recommended specifically for FMS, unless an inflammatory comorbidity is present, in which case there is an indirect effect of the NSAIDs onto the FMS pain presumably because of a reduction of the noxious load on the Central Nervous System from the peripheral inflammatory areas.

This high use should also be commented by the authors in the light of the available data from the literature.

Response 2: Lines 369-75.

We have revised the English language and style by one member of the team who is native.

And we changed the data when we did the focus group as we saw we made a mistake. They were holded in May and June not in January and February. Line 136.
